# Optimal Ventilation Design for Flammable Gas Leaking from Gas Box Used in Semiconductor Manufacturing: Case Study on Korean Semiconductor Industry

**Sang-Ryung Kim [1,*], Hyo-Shik Moon [1] and Phil-Hoon Jeong [2,*]**

[1] Korea Occupational Safety and Health Agency, Incheon 44429, Republic of Korea; hsmoon91@kosha.or.kr
[2] Department of Industrial Safety Engineering, Korea Soongsil Cyber University, Seoul 06978, Republic of Korea
[*] Correspondence: madeinhiphop@naver.com (S.-R.K.); jph@mail.kcu.ac (P.-H.J.);
    Tel.: +82-10-3500-8104 (S.-R.K.); +82-10-2519-0805 (P.-H.J.)

**Abstract:** Highly flammable substances such as hydrogen and silane are used in the semiconductor manufacturing process. When gas leaks, it is mixed with outside air and connected to a treatment facility through the duct inside the gas box. This study investigated optimal exhaust design to prevent fire explosions and health problems by optimizing the exhaust volume when hydrogen leaks from the gas box of semiconductor manufacturing equipment. After selecting the leakage rate amount based on the KS C IEC 60079-10-1, SEMI S6-0707E, and SEMI F-15 standards, a gas box was manufactured. Subsequently, the fan speed required to ventilate the gas box more than five times per minute according to the SEMI standard and the opening area and location that can reduce the lower explosive limit (LEL) to less than 25% in the event of hydrogen leakage were determined. When the air intakes were placed on the left and right, the flow rate was measured at 32 L per minute (LPM), and the maximum concentration was measured at 9111 ppm. This is less than 25% of the LEL of hydrogen and is believed to be capable of preventing fire and explosion, even if a similarly flammable gas leaks inside the gas box.

**Keywords:** gas box; SEMI S6-0707E; SEMI F-15; SEMI S6; KS C IEC 60079-10-1; tracer gas test; semiconductor manufacturing process

## 1. Introduction

The Republic of Korea is the world's top semiconductor manufacturer. Unlike other countries, residential facilities such as apartments exist close to Korean semiconductor manufacturing facilities, which necessitates the establishment of stricter safety policies compared with other countries. However, research related to semiconductor safety is difficult owing to the closed nature of the national industry, and research on effectively exhausting flammable substances when they leak from inside the gas box has only begun recently. Flammable gas leaks are one of the most difficult problems to solve in chemical-handling industries, including the semiconductor industry. In the semiconductor industry, which handles various materials, gas leaks can cause immediate damage in countries such as Korea, where residential facilities are close to industrial facilities; therefore, particular caution is required in such situations [1–3]. The Korea Occupational Safety and Health Agency, entrusted by the Ministry of Employment and Labor, is implementing a process safety management (PSM) submission system to ensure the safety of 51 chemical substance handling and related facilities, including facilities handling flammable gases. These systems are specified in the Occupational Safety and Health Act, enforcement ordinance, and enforcement regulations [4–6]. However, owing to the high complexity and risk of the semiconductor industry production process, this legal system cannot sufficiently establish standards for safety and health in the workplace. However, the Semiconductor Equipment and Materials International (SEMI) is distributing safety guidelines for reference, but the

information presented in these materials does not provide specific design standards, and the guidelines are not enforced. Consequently, semiconductor manufacturers follow different standards [7].

The semiconductor manufacturing plant shown in Figure 1 uses numerous chemicals to manufacture semiconductors. Red arrows indicate the flow of chemical liquid, and blue arrows indicate the flow of chemical gas. Important facilities that handle chemicals include the chemical supply facility (utility), the semiconductor manufacturing facility (fabrication) (FAB), and the plenum [8–10]. Among them, the gas box attached to most semiconductor manufacturing facilities (FAB, plenum) must be made of a material that does not corrode or burn when harmful or hazardous substances leak. In the event of a leak in the gas box, negative pressure must be created inside the gas box such that the pressure does not discharge to the exterior. Instruments such as MFCs and valves are connection points between facilities and can cause the leakage of harmful and hazardous substances. Therefore, appropriate exhaust facilities must exist to prevent leaks from discharging to the exterior.

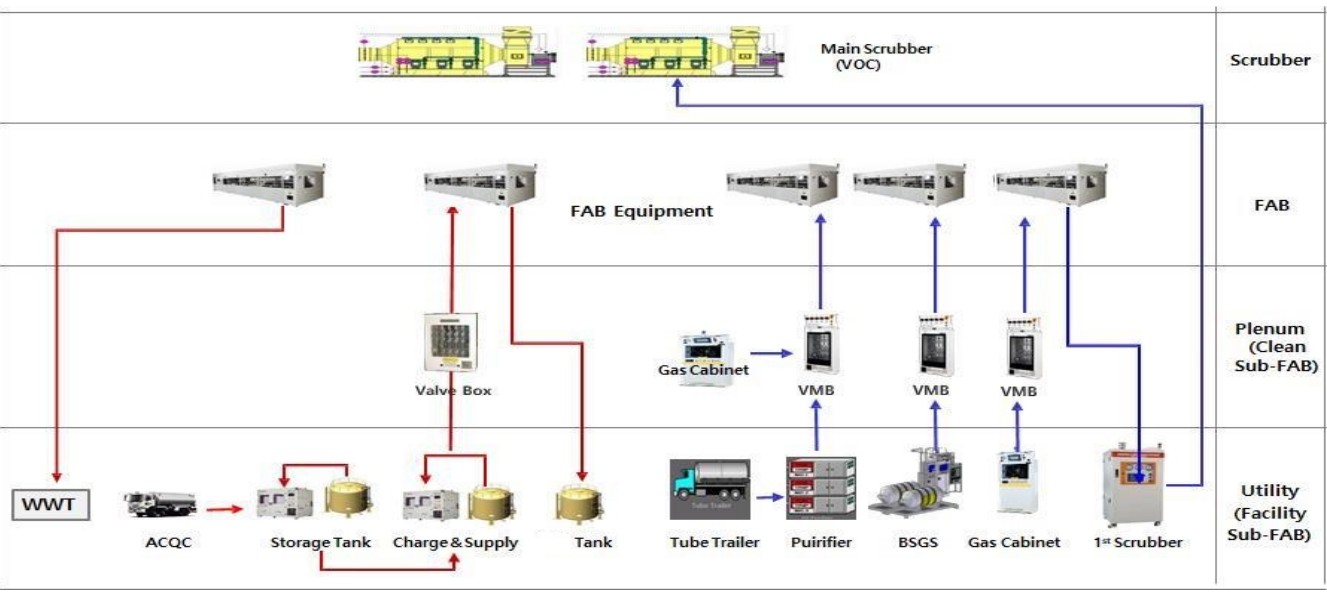

**Figure 1.** Overall semiconductor manufacturing plant flow.

In Korea, the leak rate calculation of flammable substances is based on the Korean Industrial Standard KS C IEC 60079-10-1. In addition, if the standard differential pressure designed for the exhaust is exceeded, the equipment must cease operation by generating an alarm and blocking the material supply pipe [11–13]. Particularly, gas boxes and ducts that handle spontaneous combustion substances, such as silane, must withstand temperatures above the ignition temperature. The internal structure includes a mass flow controller (MFC) installed to control the flow rate of harmful and hazardous substances in the form of gases before they go to the process chamber through the piping and a valve for maintaining it.

The hydrogen gas used in this study is extremely flammable and has a lower explosive limit (LEL) of 4%, which can cause fire or explosion. The upper explosion limit (UEL), which has a wide explosion range, is 75%. Hydrogen gas is much lighter than air; therefore, when it leaks, it easily disperses to the top. In the form of compressed gas, hydrogen is transported in cylinders, tube trailers, and storage tanks and is used in most semiconductor manufacturing processes [14,15]. The calculation basis and formula for selecting the range of an explosion hazard location are presented in KS C IEC 60079-10-1. Unlike the continuous leak rating, which applies to the entire area, and the primary leak rating, which applies to the entire leak cross-sectional area, the secondary leak rating provides guidance. Importantly, this does not clearly define the standards for calculating the leak rate but does provide guidance [16]. Also, in the remarks item, it is possible to select a method

that is already applied, such as the application of equipment manufacturer data, API RP505 or NFPA 497, or a specific country or industry code [17,18]. To select an accurate explosion hazard location, diffusion modeling software considering each operating condition for each material can be used. However, it is difficult to model all devices wherein flammable substances are used in a workplace; therefore, the application of such software is extremely limited in Korea [19]. In addition to legal standards such as KS C IEC 60079-10-1, Korean semiconductor companies comply with the Environmental, Health, and Safety Guideline for Semiconductor Manufacturing Equipment (SEMI S2) and the Environment, Health, Safety (EHS) Guideline for Exhaust Ventilation of Semiconductor Manufacturing Equipment of the Semiconductor Manufacturing Equipment and Materials International, which are implemented to obtain third-party certification for companies. These guidelines do not provide specific design standards for certain areas; therefore, semiconductor manufacturing equipment manufacturers select different emission standards based on their respective research methods to receive certification [20].

The objective of this study was to design a gas box exhaust volume that can prevent health problems for workers, even if harmful or hazardous substances leak from the gas box, by proposing design standards for the duct cross-sectional area of the gas box. In this study, the optimal exhaust method was derived by investigating the design of the duct size, air inlet requirement, and air inlet size based on the size of the gas box, according to the technical standards of SEMI S6 or industrial ventilation.

This paper is organized into four sections: (1) Introduction, (2) Experimental Setup, (3) Results, and (4) Conclusions.

## 2. Experimental Setup

### 2.1. Calculation of Release Rate of Hazardous Substances

When testing for gas box leaks, it is important to assume how much hazardous material flow is leaking through the pipes in addition to the exhaust flow rate. Gas leakage is related to stress, erosion, and electric arcs [21,22]. Because there are countless potential leak points in chemical plants, such as semiconductor plants, it is virtually impossible to prevent all gas leaks. Therefore, prompt and appropriate action is required to minimize damage resulting from gas leak accidents [23–25]

The worst-case scenario for a leak is the case wherein the pipe ruptures, and the fluid within the pipe continues to leak. However, for a semiconductor manufacturing plant, the rupture of all pipes inside the gas box is impossible unless the pipes are intentionally damaged. Regarding the leakage amount, the leakage calculation formula for hydrogen gas was obtained from KS C IEC 60079-10-1, SEMI S6-0707E, and F15. The water leakage amount was calculated using the most conservative method and the water leakage calculation formula.

### 2.1.1. KS C IEC 60079-10-1: Release Rate Equation for Hydrogen

The KS C IEC 60079-10-1 standard is based on IEC 60079-10-1 as revised in Korea. Because the revised content was published domestically, many questions have been received from the international IEC and the domestic IEC regarding definitions and expressions that had not been clearly defined, and these questions have now been resolved after several meetings [26–28].

If the internal pressure of the gas is higher than the critical pressure (Pc), the speed of the leaking gas becomes the speed of sound (choke). Here, the critical pressure is equal to Equation (1). The adopted assumptions are as follows. Considering a situation wherein corrosion may occur if hydrogen leaks from inside the gas box, the hole cross-sectional area (S) was selected as 0.25 mm$^2$ (0.25 $\times$ 10$^{-6}$ m$^2$) according to the KS C IEC 60079-10-1 guidelines. The internal pressure (P) was selected as 377,143 Pa(a), which is commonly used by semiconductor companies, and the orifice leakage coefficient ($C_d$) was selected as 0.75 with angular orifice. Furthermore, the atmospheric pressure was 101,325 Pa, the ideal gas constant was 8314 J/kmol·k, the absolute temperature was 293 K, the polytropic index

was 1.41, and the hydrogen molecular weight used was 2 kg/kmol. When computed using this method, the internal pressure of gas exceeds the critical pressure, and Equation (2) is used in this case. The factors required for calculation are presented in Table 1.

$$P_C = P_a \left(\frac{r+1}{2}\right)^{\frac{r}{r-1}} \tag{1}$$

$$W_g = C_d SP \sqrt{\frac{MR}{ZRT} \left(\frac{2}{r+1}\right)^{(r+1)/(r-1)}} \tag{2}$$

**Table 1.** Factors required for KS C IEC 60079-10-1 calculation.

| Sign | Meaning |
| --- | --- |
| $P_C$ | Critical pressure (Pa) |
| $P_a$ | Atmospheric pressure (Pa) |
| P | Internal pressure (Pa) |
| $\gamma$ | Polytropic index (dimensionless) |
| $C_d$ | Discharge coefficient (dimensionless) |
| $Wg$ | Mass leak rate (kg/s) |
| R | Ideal gas constant (8314 J/kmol·k) |
| S | Hole cross-sectional area $(mm^2)$ |
| Z | Compressibility factor (dimensionless) |
| T | Absolute temperature (K) |
| M | Molecular weight (kg/kmol) |

2.1.2. SEMI S6-0707E: Release Rate Equation for Hydrogen

The SEMI S6 calculation formula essentially assumes that the pipe has ruptured [5]. To calculate this, Equations (3)–(6) are used. The factors required for calculation are presented in Table 2. In this equation, the pipe length (L) is the denominator of the leakage amount; therefore, as the length increases, the amount of leakage decreases. Because there is no standard for the pipe length, a conservative approach was used to calculate the leakage amount: the length of the pipe was considered as 1 m based on the box height (1 m), and the diameter was considered as 0.00635 m, which is mainly used in gas boxes employed by the semiconductor companies. The density of gas flowing through a straight tube downstream was 0.0000835 g/cm$^3$, the upstream absolute pressure was 377,143 Pa(a), and the hydrogen molecular weight used was 2 kg/kmol.

$$rM_1^2 = \frac{1 - \left[\frac{P_0}{P_1}\right]^2}{4f\left[\frac{L}{D}\right] + \ln\left[\frac{P_1}{P_0}\right]^2} \tag{3}$$

$$\rho_1 = \rho_0 \left[\frac{P_1}{P_0}\right] \tag{4}$$

$$w = 0.00001SS\left[0.1\rho_1 P_1 rM_1^2\right]^{0.5} \tag{5}$$

$$Q = 60\frac{wV}{M} \tag{6}$$

**Table 2.** Factors required for SEMI S6-0707E calculation.

| Sign | Meaning |
|---|---|
| $\rho_0$ | Density of gas flowing through straight tube at downstream (ambient) condition $(g/cm^3)$ |
| $\rho_1$ | Density of gas flowing through straight tube at upstream condition $(g/cm^3)$ |
| $4f$ | 0.02 surface roughness parameter for smooth pipe (dimensionless) |
| $\gamma$ | Polytropic index (dimensionless) |
| $w$ | Mass flow rate of gas flowing in straight tube (kg/s) |
| $M_1^2$ | Square of upstream Mach number (dimensionless) |
| $P_1$ | Upstream absolute pressure (pa) |
| $P_0$ | Downstream absolute pressure (pa) |
| L | Pipe length (m) |
| D | Pipe diameter (m) |
| Q | Volume flow rate of gas (L/min) |
| V | 22.4 (L/mole) molar ideal gas volume |
| M | Molecular weight (kg/kmol) |
| S | Hole cross $-$ sectional area $(mm^2)$ |

2.1.3. SEMI F-15: Release Rate Equation for Hydrogen

If there is no guideline regarding a particular discharge volume, 28 SLPM is defined for 0.00635 m piping according to SEMI F-15, widely used in the SEMI S6 third-party certification [11,29,30].

2.1.4. Release Rate Calculation Result for Hydrogen

The pipe pressure and leakage hole cross-sectional area are important variables in calculating leakage volume. The maximum pressure inside the gas box is mostly 377,143 Pa(a); therefore, this value was used. In the case of the leakage hole cross-sectional area, it is almost impossible to assume that the pipe inside the gas box will rupture, and determining the exhaust volume based on this leak amount will result in a very inefficient design. Therefore, the experiment was based on a flow rate of 32 LPM that can leak during normal operation. The calculation results are presented in Table 3.

**Table 3.** Release rate calculation results.

| Calculation Source | Pressure (Pa(a)) | Release Opening $(mm^2)$ | Volume Flow Rate of Gas (Liters per Minute) |
|---|---|---|---|
| KS C IEC 60079-10-1 | 377,143 | 0.25 | 32 |
| SEMI S6-0707E | 377,143 | 31.65 | 3146 |
| SEMI F-15 | 377,143 | - | 28 |

*2.2. Gas Box Exhaust System Design Criteria and Selection of Experimental Gas*

The experiments were performed based on the selected leakage rate of 32 LPM. Because it is very dangerous to use actual hydrogen gas when testing for leaks inside the gas box, an inert gas such as SF6 is diluted with nitrogen gas and used as a tracer gas. Then, samples are collected from inside the gas box and analyzed [31,32]. For the analyzed sample, the equivalent emission concentration is calculated according to the following equation. The concentration is calculated using Equation (7), and the experimental equipment is shown in Figure 2. The basic exhaust conditions for the experiment are as follows.

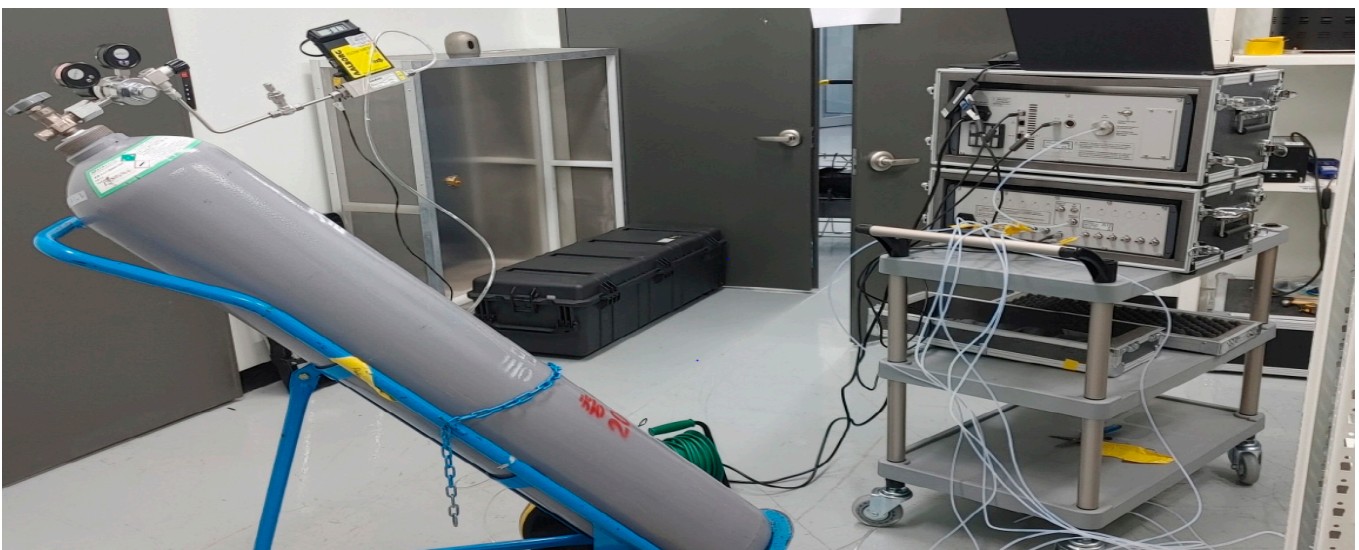

**Figure 2.** Tracer gas (SF6 1%, 99% $N_2$) and photoacoustic gas monitor.

① Air replacement should be possible at least 4–5 times per minute for the rapid dilution of the explosive atmosphere;
② The duct transfer speed should be 5 m/s or more, such that leaked flammable gases are quickly removed;
③ Ensure that the flammable gas concentration is within 25% of the LEL within a short period after leakage.

$$\text{Equivalent Release Concentration(ERC)} = (\text{Measured tracer gas concentration}) \times (\text{Process gas concentration}) / (\text{Injected tracer gas concentration}) \tag{7}$$

### 2.3. Gas Box Modeling

When installing a local exhaust system in a location where pollutants are consistently leaking, at least one side must be opened. This ensures that sufficient air enters the hood to dilute the pollutants. Additionally, the exhaust system must be connected to an air purification device.

Because there are currently no standards for air inlets in the gas boxes of semiconductor manufacturing equipment, in some cases, all gas box air inlets are blocked, and instead of air inlets, pipe inlets are made larger than the pipe diameter to replace the air inlets. Recently, all gas pipe inlets began to be blocked after a gas box leak incident. In theory, if there is no air inlet, a vacuum is maintained in the gas box. Therefore, even if the difference between the internal and external pressure is high, the airflow is expected to be low, and harmful or hazardous substances are not diluted. Therefore, an air inlet with an appropriate size is required.

The gas box in this study was selected as the gas box used in the recently developed etching process. The selected equipment is being developed to carry out an additional chemical vapor deposition (CVD) process to compensate for film defects that occur after etching, in addition to etching. Therefore, the gas box was manufactured with a larger size compared with that of existing etching equipment. Specifically, the size of the gas box was 600 mm (0.6 m) × 350 mm (0.35 m) × 1000 mm (1 m). An air inlet with a width of 10 mm (0.01 m) and height of 125 mm (0.125 m) was installed on the front of the gas box, as shown in Figure 3.

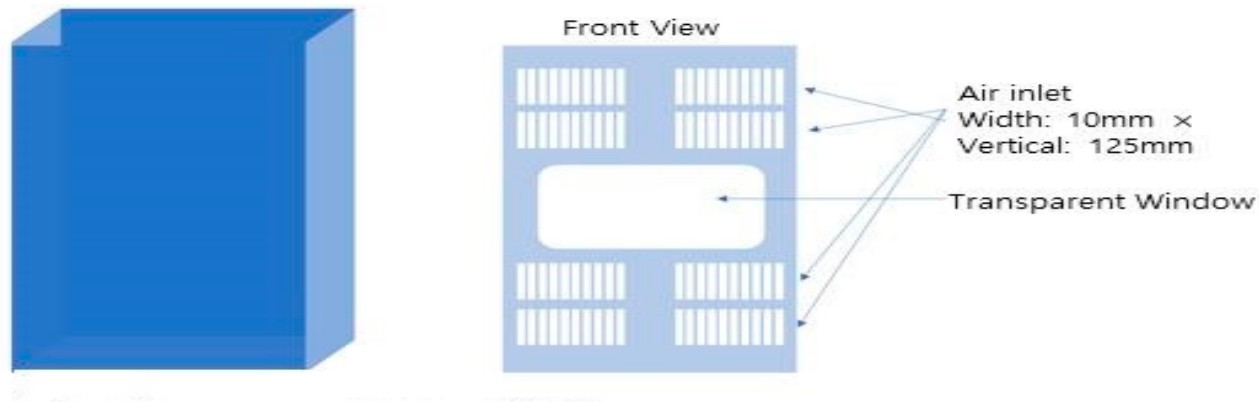

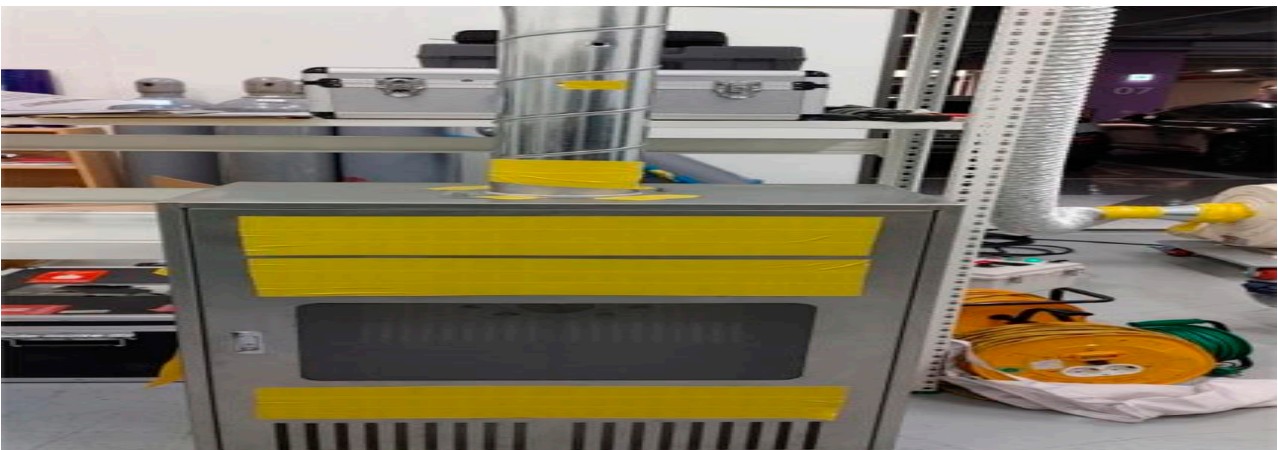

**Figure 3.** Gas box modeling.

## 2.4. Duct Diameter Selection

① According to the basic exhaust conditions for the experiment discussed in Section 2.2, a 150 mm (0.15 m) duct was connected, and the fan speed was adjusted to allow ventilation to occur more than five times. The duct wind speed was measured by opening each air inlet (width: 10 mm (0.01 m), height: 12 mm (0.012 m)) at the bottom of the gas box one by one. The wind volume was calculated from the measured wind speed, and the number of ventilation times per hour was confirmed;

② A 75 mm (0.075 m) duct was connected, and the fan speed was adjusted to allow ventilation to occur more than five times. This time, because the cross-sectional area of the duct was 25% less than 150 mm (0.15 m), the air inlet (width: 10 mm (0.01 m), height: 12 mm (0.012 m)) at the bottom of the gas box was opened to 50% compared with the 150 mm (0.15 m) duct, and the wind speed was measured. The wind volume was calculated from the measured wind speed, and the number of ventilation times per hour was confirmed;

③ To remove leaked flammable gases quickly, a diameter that allowed a duct transport speed greater than 5 m/s was selected.

## 2.5. Air Inlet Location Selection

After selecting the duct diameter as discussed in Section 2.4, the air inlet position was adjusted to determine the optimal opening location by measuring the point where the combustible gas concentration at the sampling point was within 25% of the lower explosion. The sampling points inside the gas box are shown in Figure 4.

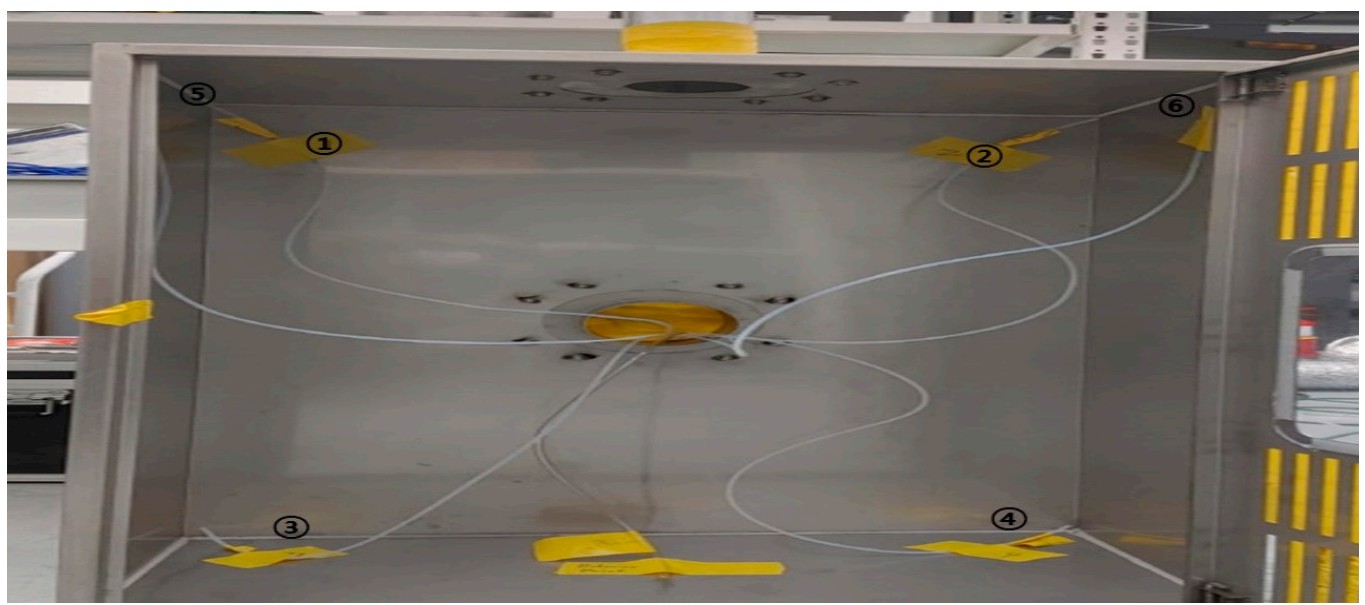

**Figure 4.** Gas box modeling sampling point in gas box (S: sampling point 1–6).

### 3. Results

*3.1. Duct Transfer Speed Results According to Duct Size*

3.1.1. Results for 150 mm (0.15 m) Duct Experiment

　　The results of the 150 mm (0.15 m) duct test reveal that the differential pressure had the highest value when all air inlets were closed. As the air inlets opened, the pressure in the duct rapidly decreased, but the wind speed and volume did not increase significantly above 14–21% of the duct cross-sectional area. Even when the number of ventilations per minute of the gas box was more than 10, the speed was 2 m/s, which does not satisfy the minimum duct transfer speed of 5 m/s for gaseous materials. The 150 mm (0.15 m) duct was not applied because it did not satisfy the experimental standards. The experimental results are summarized in Table 4.

**Table 4.** Experimental results for 150 mm (0.15 m) duct.

| Air Inlet | | | | | | |
| Horizontal Length (mm) | Vertical Length (mm) | Number of Air Inlets | Opening Area (mm²) | Internal Conveyance Speed of Duct (m/s) | Number of Ventilation Times per Minute | Differential Pressure (Pa) |
|---|---|---|---|---|---|---|
| 10 | 125 | 0 | 0 | 1 | 5.05 | 65 |
| 10 | 125 | 1 | 1250 | 1.5 | 7.57 | 57 |
| 10 | 125 | 2 | 1250 | 1.8 | 9.09 | 40 |
| 10 | 125 | 3 | 3750 | 2 | 10.10 | 26 |
| 10 | 125 | 4 | 5000 | 2.1 | 10.60 | 18.6 |
| 10 | 125 | 5 | 6250 | 2.15 | 10.86 | 13.7 |
| 10 | 125 | 6 | 7500 | 2.2 | 11.11 | 10 |
| 10 | 125 | 7 | 8750 | 2.25 | 11.36 | 8 |

3.1.2. Experimental Results for 75 mm (0.75 m) Duct

　　As with the 150 mm (0.15 m) duct, the differential pressure was highest when all air intakes were closed. As the air intakes opened, the differential pressure inside the duct rapidly decreased. In the 75 mm (0.075 m) duct, even when only one air intake

port was opened, the duct speed was maintained at an appropriate transport speed of 6.4 m/s. Currently, most semiconductor manufacturing equipment companies assess exhaust adequacy based on the differential pressure of the duct, and approximately 100 Pa satisfies the standards of most semiconductor equipment companies.

As a result of determining the adequacy of the ratio of the air inlet to the duct area using a 75 mm (0.075 m) duct connection experiment, when the air inlet area was opened by approximately 57% of the duct cross-sectional area, the number of ventilation times per minute was more than eight, and the differential pressure in the duct was approximately 100 Pa. Therefore, the ratio of the air inlet to the duct area was assessed to be appropriate. The experimental results are summarized in Table 5.

**Table 5.** Experiment results for 75 mm (0.75 m) duct.

| Air Inlet | | | Opening Area (mm²) | Internal Conveyance Speed of Duct (m/s) | Number of Ventilation Times per Minute | Differential Pressure (Pa) |
| --- | --- | --- | --- | --- | --- | --- |
| Horizontal Length (m) | Vertical Length (m) | Number of Air Inlets | | | | |
| 10 | 125 | 0 | 0 | 4.1 | 5.18 | 192 |
| 10 | 125 | 0.5 | 625 | 4.9 | 6.19 | 150 |
| 10 | 125 | 1 | 1250 | 6.4 | 8.08 | 146 |
| 10 | 125 | 1.5 | 1875 | 6.8 | 8.58 | 138 |
| 10 | 125 | 2 | 2500 | 7 | 8.84 | 98 |
| 10 | 125 | 2.5 | 3125 | 7.5 | 9.47 | 81 |
| 10 | 125 | 3 | 3750 | 8 | 10.10 | 70 |
| 10 | 125 | 3.5 | 4375 | 8 | 10.10 | 70 |

*3.2. Flammable Gas Concentration Measurement Results According to Air Inlet Location*

Based on the 75 mm (0.075 m) diameter duct discussed in Section 3.2, 0% and 57% of the duct cross-sectional area was opened, tracer gas (SF6 1% + N2 99%) was released into the gas box at 32 LPM, and the internal concentration was measured.

3.2.1. Concentration inside Gas Box with Opening Area of 0% of Duct and Differential Pressure of 192 Pa

By measuring the concentration with all openings blocked, it was found that the concentration exceeded 25% (10,000 ppm) of the LEL at all sampling points, which does not satisfy the safety standards. Particularly, points 3 and 4, which are close to the leak point, were 21,133 ppm, which is more than double the 25% (10,000 ppm) of the LEL.

Based on theory, it is expected that if the air inlet is blocked, a vacuum will be created, and there will be no airflow. In reality, however, the airtightness of the box is not ideal; therefore, air flows in and forms a small flow rate. A gas box with an opening area of 0% is shown in Figure 5, and the experimental results are presented in Table 6.

The ERC of the process chemicals can be calculated using the tracer gas concentration measured with a multi-gas detector. The ERC was compared to the LEL of the process chemical, and an ERC lower than 25% of the LEL was considered acceptable.

In the case of S1, the measurement sample, the measured SF6 concentration was 119.41 ppm. The ERC value can be obtained using Equation (7).

$$\text{ERC} = 119.41 \text{ ppm} \times 100/1\% = 11{,}941 \text{ ppm}$$

$$\text{LEL of H2} = 40{,}000 \text{ ppm}$$

$$\text{ERC/LEL (\%)} = 11{,}941 \text{ ppm}/40{,}000 \text{ ppm} \times 100\% = 29.85\%$$

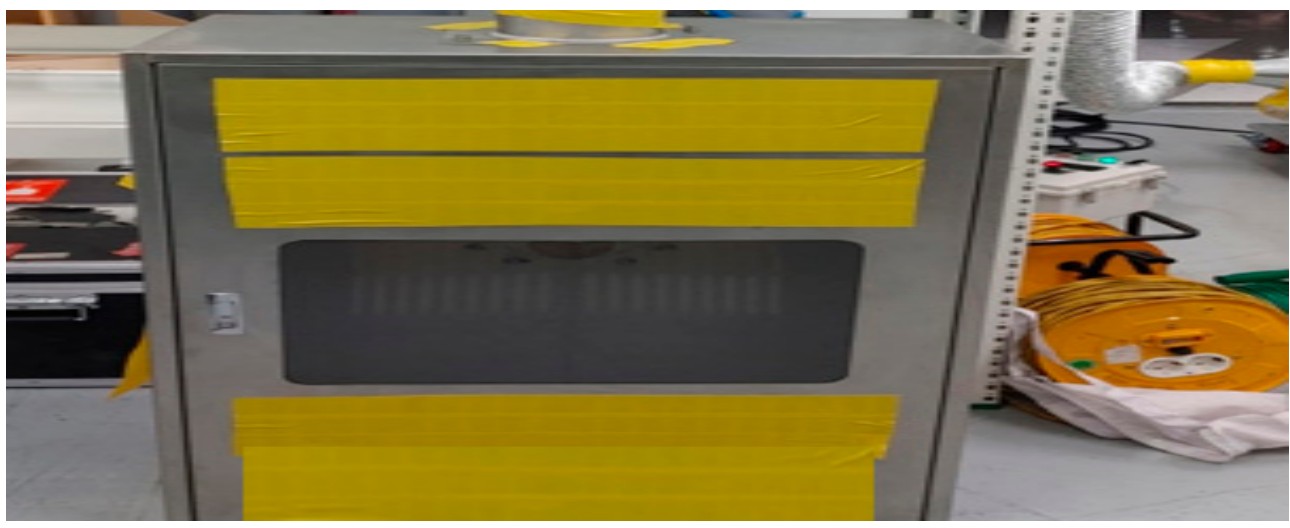

**Figure 5.** Gas box modeling with 0% opening area.

**Table 6.** Sampling point concentration (0% opening area).

| Sampling Point | S1 | S2 | S3 | S4 | S5 | S6 |
|---|---|---|---|---|---|---|
| Measured value (ppm) | 119.41 | 129.41 | 211.33 | 200.43 | 109.22 | 134.89 |
| Equivalent concentration (ppm) | 11,941 | 12,941 | 21,333 | 20,043 | 10,922 | 13,489 |
| Reference concentration (ppm) | 10,000 | 10,000 | 10,000 | 10,000 | 10,000 | 10,000 |
| % LEL | 29.85 | 32.35 | 52.83 | 50.11 | 27.30 | 33.72 |
| Pass/Fail | Fail | Fail | Fail | Fail | Fail | Fail |

3.2.2. Concentration inside Gas Box with 57% Duct Opening Area and Differential Pressure of 98 Pa

By measuring the concentration with an opening area of 57% of the duct cross-sectional area, it was found that, in some cases, the concentration exceeded 10,000 ppm, which is 25% of the LEL. The gas box with an opening area of 57% is shown in Figure 6, and the experimental results are presented in Table 7.

**Table 7.** Sampling point concentration (57% opening area).

| Sampling Point | S1 | S2 | S3 | S4 | S5 | S6 |
|---|---|---|---|---|---|---|
| Measured value (ppm) | 81.27 | 105.76 | 102.52 | 69.39 | 82.10 | 104.64 |
| Equivalent concentration (ppm) | 8127 | 10,576 | 10,252 | 6939 | 8210 | 10,464 |
| Reference concentration (ppm) | 10,000 | 10,000 | 10,000 | 10,000 | 10,000 | 10,000 |
| % LEL | 20.32 | 26.44 | 25.63 | 17.35 | 20.53 | 26.16 |
| Pass/Fail | Pass | Fail | Fail | Pass | Pass | Fail |

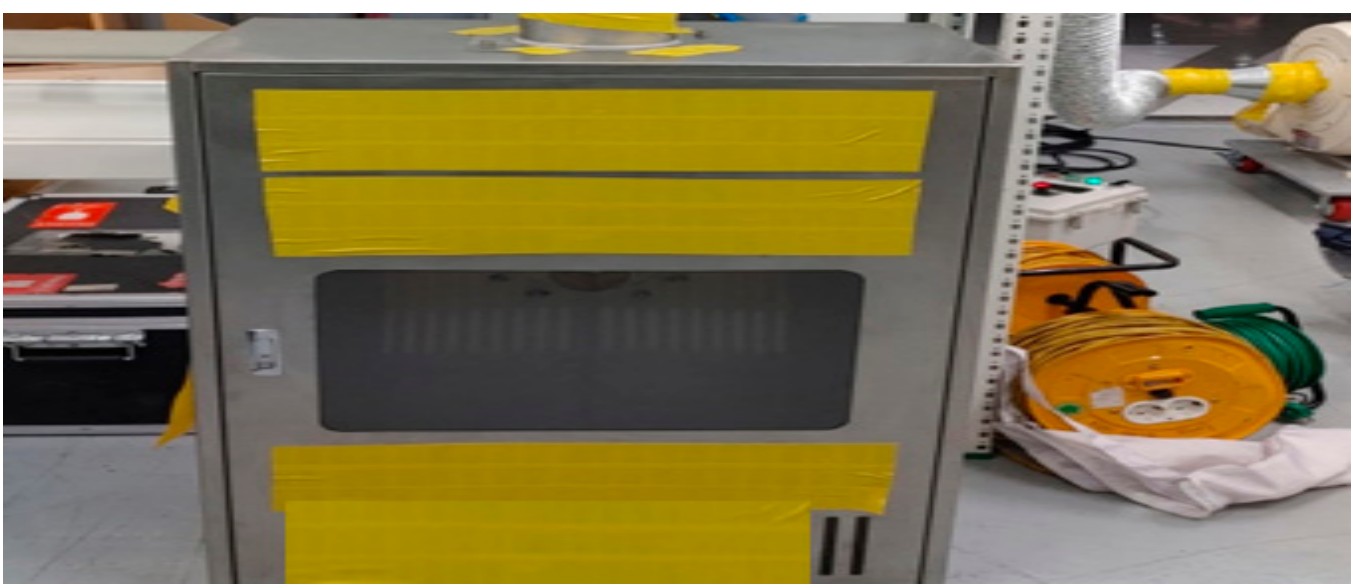

**Figure 6.** Gas box modeling with 57% opening area.

### 3.2.3. Concentration inside Gas Box with 57% Duct Opening Area (Both Openings) and Differential Pressure of 98 Pa

Table 8 presents the results of separating the air intake into left and right, as shown in Figure 7. Under the same conditions as the experiment discussed in Section 3.2.2, areas 3 and 4 around the air inlet had relatively low concentrations, and the top of the box also maintained 20% of the LEL. Dead zones did not occur when one side of the air intake was opened.

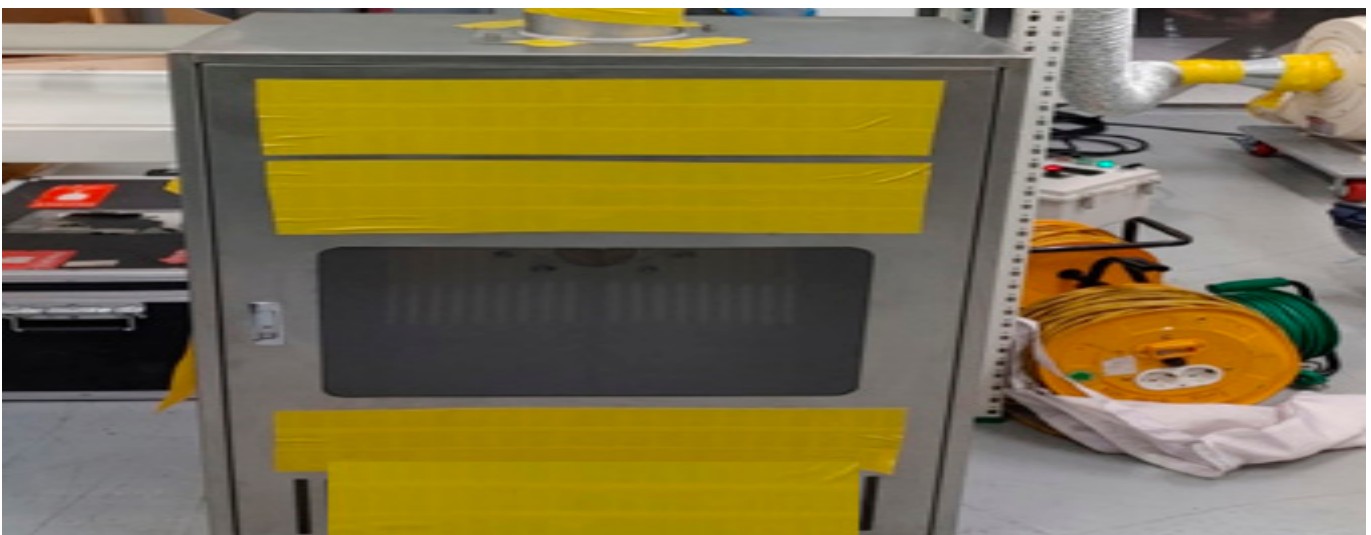

**Figure 7.** Gas box modeling with 57% opening area (both openings).

**Table 8.** Sampling point concentration (57% opening area, both openings).

| Sampling Point | S1 | S2 | S3 | S4 | S5 | S6 |
|---|---|---|---|---|---|---|
| Measured value (ppm) | 81.95 | 90.90 | 77.18 | 71.42 | 80.35 | 91.11 |
| Equivalent concentration (ppm) | 8195 | 9090 | 7718 | 7142 | 8035 | 9111 |

**Table 8.** *Cont.*

| Sampling Point | S1 | S2 | S3 | S4 | S5 | S6 |
|---|---|---|---|---|---|---|
| Reference concentration (ppm) | 10,000 | 10,000 | 10,000 | 10,000 | 10,000 | 10,000 |
| % LEL | 20.49 | 22.73 | 19.30 | 17.86 | 20.09 | 22.78 |
| Pass/Fail | Pass | Pass | Pass | Pass | Pass | Pass |

## 4. Conclusions

Although Korea is one of the world's major producers of semiconductor products, there is not much information available regarding the safety and health standards that must be followed during semiconductor manufacturing. The exhaust system is one of the main safety and health facilities in semiconductor manufacturing. The gas box serves as a safety device to ensure the safety and health of the workplace by limiting the spread of gas and discharging it through the exhaust port if harmful or dangerous substances leak from the piping of the semiconductor manufacturing equipment. The connection of the gas box to the exhaust system is also essential for preventing health problems for workers in the event of hazardous substance leakage. Nevertheless, because safety and health design guidelines have not been established at semiconductor plants, the installation of gas boxes does not follow specific standards. This study investigated how to maintain optimal air volume to prevent the exposure of workers to health hazards while minimizing the energy consumption of exhaust fans. To investigate the optimal exhaust method for the gas box of semiconductor manufacturing equipment, a 0.21 m$^3$ gas box (size of 600 mm (0.6 m) × 350 mm (0.35 m) × 1000 mm (1 m)) was manufactured. After replacing the interior more than five times and selecting a duct size (75 mm (0.075 m)) that could allow the duct's internal conveyance speed to be higher than 5 m/s, the concentration inside the gas box was measured using flowing tracer gas at 32 LPM.

When 32 LPM was leaked into the gas box without an air opening area, the concentration inside the box was measured at a maximum of 21,133 ppm. Therefore, even flammable gases such as hydrogen cannot be considered safe because they exceed 10,000 ppm, which is less than 25% of the LEL. Particularly, acetylene, which has a LEL of less than 1%, can produce an explosion even if it is exhausted.

When there is an air inlet that was opened only on one side of the bottom of the gas box, inside the gas box, a minimum of 6939 ppm and a maximum of 10,576 ppm were measured based on the equivalent release concentration. When one side of the air inlet was opened, a dead zone occurred inside the gas box, which did not satisfy the safety standards for fire and explosion prevention. The experimental results confirm that an air inlet must be present and that a dead zone may occur if the air inlet location is biased. When the air intake was placed on both the left and right sides, the maximum concentration was measured at 9111 ppm, which is considered safe according to hydrogen standards.

The experiments in this study were not based on an empirical model used in semiconductor manufacturing plants. Moreover, the exhaust fluid resistance caused by the piping and valves inside the gas box was not reflected in this study. Despite these limitations, the research is meaningful because it establishes a procedure for designing the gas box exhaust and confirms the size and location standards for the duct cross-sectional area and air inlet. For semiconductor manufacturing equipment manufacturers and manufacturing sites, gas boxes and gas box exhausts should be implemented as safety devices to lower the risk of fire and explosion and prevent the exposure of workers to harmful and hazardous substances. To this end, equipment manufacturers and equipment operators at manufacturing sites should check the design standards for gas boxes and use and maintain these standards accordingly.

**Author Contributions:** Conceptualization, S.-R.K.; methodology, S.-R.K.; validation, P.-H.J.; writing—original draft preparation, S.-R.K. and P.-H.J.; writing review, H.-S.M.; editing, S.-R.K. All authors have read and agreed to the published version of the manuscript.

**Funding:** This research received no external funding.

**Institutional Review Board Statement:** Not applicable.

**Informed Consent Statement:** Not applicable.

**Data Availability Statement:** Some or all data used in this research are available from the corresponding author upon request.

**Conflicts of Interest:** The authors declare no conflict of interest.

## Abbreaviations (Alphabetical Order)

CVD     Chemical Vapor Deposition
ERC     Equivalent Release Concentration
EHS     Environmental, Health, Safety
FAB     Fabrication
LEL     Low Explosion Limit
LPM     Liters Per Minute
SEMI     Semiconductor Equipment and Materials International
UEL     Upper Explosion Limit

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
