# Peer review of "Optimal Ventilation Design for Flammable Gas Leaking from Gas Box Used in Semiconductor Manufacturing: Case Study on Korean Semiconductor Industry"

_fire, doi:10.3390/fire6110432_

Round 1

Reviewer 1 Report

Comments and Suggestions for Authors

This manuscript presents the design method for a gas box exhaust volume (in terms of the duct size, air inlet requirements, air inlet size, etc.) to prevent health problems for workers when harmful or hazardous substances leak from the gas box. The manuscript fits the scope of Fire but needs major revision. My detailed comments are given below.

(1) For better understanding of the research object please provide a schematic of the gas box with all notations used in the manuscript.

(2) Before formulating the calculation approach, please provide the list of all adopted assumptions.

(3) Please provide the Nomenclature of all notations (some variables like Z in Eq.(1), f in Eq.(3), etc. are not defined in the text).

(4) Please provide all dimensional values in the SI system (Système international d'unités). There is a mix of units in the text: length is measured either in inches or in mm/cm/m, pressure in psi or Pa, etc.

(5) Please note that Eq. (1) contains errors, Eq. (4) has a typo.

(6) Please use a single notation for one variable like r in Eq. (1) and k in Eq. (3) or like M in Eq.(2) and M_gas in Eq. (6), or S in Eq. (2) and A in Eq. (5).

(7) The numbers assumed in lines 167-169 must be grounded. Please provide a reference.

(8) Please use the same units for the same variables in the manuscript. Thus, variable Q is measured in liter/sec (Table 1), whereas in Table 2 it is measured in liters/min; opening area is measured in mm^2 in Table 2 and in m^2 in Table 3; pressure is measured in psi in Table 2 and in Pa in Table 3.

(9) Equation (7) is first mentioned in Line 189 and must appear in the next line rather than in Line 197.

(10) Please provide solid grounds for the possibility of modeling hydrogen leak by the leak of SF6. These substances differ drastically in all physical properties entering the governing equations including molecular weight, the ratio of specific heats, and viscosity. Despite the amount of added SF6 in your experiments does not exceed 1 vol%, the behavior of H2 and SF6 in the gas box can be different. Please provide more information on how the gases mix in the gas box with air, etc. Is it guaranteed that both gases (H2 and SF6) are completely mixed with air inside the gas box? Please provide an additional schematic of the expected flow pattern inside the gas box to show the flow directions of inlet air and leaked gas.

(11) Please indicate the "new" and "old" air inlets in Figure 3.

(12) Please indicate the units of length in the first two columns of Tables 3 and 4 (mm, cm, m ?).

(13) Lines 314 and 316: H2 must be replaced by SF6.

Comments on the Quality of English Language

English is acceptable

Author Response

We appreciate the referee’s valuable time and comments. We revised the manuscript according to the referee’s comments. The revised phrases or sentences are highlighted in the revised manuscript and in this response letter.

It was uploaded as a word file.

Reviewer 2 Report

Comments and Suggestions for Authors

The article has a clear topic selection, clear expression, reasonable structure, and sufficient discussion. The article can be accepted.

Author Response

We appreciate the referee’s valuable time and comments. In the future, we will focus more on research and strive to submit excellent articles. I would like to express my sincere gratitude once again for the referee's opinion.

Reviewer 3 Report

Comments and Suggestions for Authors

Title:  

Optimal ventilation design for flammable gas leaking from gas 2 box used in semiconductor manufacturing: case study on Korean semiconductor industry.

Originality: 

- optimal exhaust design

- manufactured gas box

- lowering the LEL to less than 25%

Abstract:

No reviewer comments.

Introduction:

No reviewer comments.

Experimental Setup:

p. 5, r. 186 “…an inert gas such as SF6 is diluted with nitrogen gas and used as a tracer gas.”

Note: Sulfur hexafluoride even if diluted with nitrogen is much heavier and bigger molecule than H2 molecule. There is a question of the suitability of such leakage measurement.  

How was the hydrogen homogenized within the vessel?

Results and Discussion:

p. 8-12

The authors focus mainly on the description of the results rather than on the interpretation and discussions.

Conclusions:

p. 12, r. 362, p. 13, r. 367 etc.

Is it possible to evaluate the uncertainty of the results?

After reading the manuscript reviewer find that the article is within the scope of Fire journal and suggest to accepted the manuscript after major revisions described above.

Comments on the Quality of English Language

Title:  

Optimal ventilation design for flammable gas leaking from gas 2 box used in semiconductor manufacturing: case study on Korean semiconductor industry.

Originality: 

- optimal exhaust design

- manufactured gas box

- lowering the LEL to less than 25%

Abstract:

No reviewer comments.

Introduction:

No reviewer comments.

Experimental Setup:

p. 5, r. 186 “…an inert gas such as SF6 is diluted with nitrogen gas and used as a tracer gas.”

Note: Sulfur hexafluoride even if diluted with nitrogen is much heavier and bigger molecule than H2 molecule. There is a question of the suitability of such leakage measurement.  

How was the hydrogen homogenized within the vessel?

Results and Discussion:

p. 8-12

The authors focus mainly on the description of the results rather than on the interpretation and discussions.

Conclusions:

p. 12, r. 362, p. 13, r. 367 etc.

Is it possible to evaluate the uncertainty of the results?

After reading the manuscript reviewer find that the article is within the scope of Fire journal and suggest to accepted the manuscript after major revisions described above.

Author Response

We appreciate the referee’s valuable time and comments. We revised the manuscript according to the referee’s comments. The revised phrases or sentences are highlighted in the revised manuscript and in this response letter.

it was uploaded as a word file.

Round 2

Reviewer 1 Report

Comments and Suggestions for Authors

The authors have addressed all my comments properly. The manuscript could be now considered for publication as it is.

Reviewer 3 Report

Comments and Suggestions for Authors

Authors satisfactory answer to all reviewer comments. Therefore, I suggest a paper for publication in the Fire Journal of MDPI.